# *Galleria mellonella* Model of Coccidioidomycosis for Drug Susceptibility Tests and Virulence Factor Identification

**DOI:** 10.3390/jof10020131

**Published:** 2024-02-05

**Authors:** Matthew Mendoza Barker, Sarah Saeger, Althea Campuzano, Jieh-Juen Yu, Chiung-Yu Hung

**Affiliations:** Department of Molecular Microbiology and Immunology, South Texas Center for Emerging Infectious Diseases, The University of Texas at San Antonio, San Antonio, TX 78249, USA; matthew.barker@utsa.edu (M.M.B.); sarah.saeger@utsa.edu (S.S.); althea.campuzano@utsa.edu (A.C.); jiehjuen.yu@utsa.edu (J.-J.Y.)

**Keywords:** *Coccidioides*, Valley Fever, fungi, *Galleria mellonella*, drug susceptibility, virulence factors, mini-host

## Abstract

Coccidioidomycosis (CM) can manifest as respiratory and disseminated diseases that are caused by dimorphic fungal pathogens, such as *Coccidioides* species. The inhaled arthroconidia generated during the saprobic growth phase convert into multinucleated spherules in the lungs to complete the parasitic lifecycle. Research on coccidioidal virulence and pathogenesis primarily employs murine models typically associated with low lethal doses (LD_100_ < 100 spores). However, the *Galleria* model has recently garnered attention due to its immune system bearing both structural and functional similarities to the innate system of mammals. Our findings indicate that *Coccidioides posadasii* can convert and complete the parasitic cycle within the hemocoel of the *Galleria* larva. In *Galleria*, the LD_100_ is between 0.5 and 1.0 × 10^6^ viable spores for the clinical isolate *Coccidioides posadasii* C735. Furthermore, we demonstrated the suitability of this model for in vivo antifungal susceptibility tests to validate the bioreactivity of newly discovered antifungals against *Coccidioides*. Additionally, we utilized this larva model to screen a *Coccidioides posadasii* mutant library showing attenuated virulence. Similarly, the identified attenuated coccidioidal mutants displayed a loss of virulence in a commonly used murine model of coccidioidomycosis. In this study, we demonstrated that *Galleria* larvae can be applied as a model for studying *Coccidioides* infection.

## 1. Introduction

Coccidioidomycosis (CM) or San Joaquin Valley Fever (VF) is a fungal infection caused by *Coccidioides immitis* and *Coccidioides posadasii* and is predominantly found in endemic areas, most notably the southwest of the United States, Mexico, and arid regions of South America [1,2]. It is estimated that about 350,000 people in the US become infected with *Coccidioides* every year [3]. Valley Fever manifests as acute or progressive pneumonia, extrapulmonary non-meningeal mycosis, or meningitis [4,5]. *Coccidioides* spp. have a unique dimorphic life cycle characterized by the conversion of saprobic-phase arthroconidia into those of the parasitic spherule phase. The fungi grow as mycelia in the soil and produce arthroconidia (spores) as they mature. When the contaminated soil is disturbed, arthroconidia are released into the air, where they may be inhaled by a susceptible host. Following inhalation, arthroconidia round up and become immature spherules via isotropic growth. The spherules continue to mature via further circumferential organism growth and undergo multiple rounds of nuclear division. Eventually, the spherules become filled with hundreds of endospores (2–4 µm in diameter) after endogenous cell wall growth and cytoplasmic compartmentalization. When a mature spherule ruptures, those endospores are released, and each has the potential to form another spherule. The ability to form spherules from arthroconidia is required for coccidioidal pathogenicity [6,7]. The identification and validation of coccidioidal virulence factors associated with pathogenesis have slowly progressed since the development of molecular tools for coccidioidal gene deletion and complementation two decades ago [8,9,10,11,12,13]. The slow progression of coccidioidal virulence factor discovery is due to multiple factors, including the complexity of the life cycles of a multinucleated organism, the limitation of robust genetic tools and in vivo screening models, and the requirement of BSL3 containment for *Coccidioides* research.

There are currently no effective vaccines against this fungal infection, and the currently available antifungal medicines are limited to the polyene Amphotericin B and azoles, which are fungistatic. Long-term use of these medications can lead to hepatoxic and nephrotoxic side effects [14]. Moreover, azole-resistant strains of *Coccidioides* are emerging [15]. The increasing health- and cost-related impacts of coccidioidomycosis are a major public health challenge and urgently necessitate the development of novel antifungal drugs for this disease. High-throughput antifungal screening platforms developed for other medically important fungal pathogens can be easily modified to screen various drug-like compound libraries to identify potential anti-*Coccidioides* compounds. However, the bottleneck of antifungal development is the down-selection and prioritization of drug candidates obtained from library screenings using a robust in vivo drug efficacy evaluation system. The murine infection model is widely used to study the virulence mechanisms and immunity of *Coccidioides* infection due to the many genetically modified strains available to researchers [16,17,18,19]. However, ethical, budgetary, and BSL3 logistical hurdles are associated with using mice as a rapid evaluation model. Therefore, there is a need to establish an alternative model for studies of coccidioidal virulence and infection that can evaluate anti-*Coccidioides* drug candidates. One such potential model is a model of larvae of *Galleria mellonella*, the greater wax moth [20]. The larva model has become popular recently for its benefits in studying virulence factors of bacterial and fungal pathogens [20,21,22]. Its similarity in terms of innate immunity to that of vertebrates and its ability to be maintained at the physiological body temperature (37–40 °C) make *G. mellonella* a suitable surrogate model for microbial infection studies [20,23]. *Galleria* has been established as an infection model for several medically important fungi, including *Candida* spp. [24,25], *Aspergillus* spp. [26], *Mucoromycetes* [27], *Histoplasma capsulatum* [28], and *Cryptococcus neoformans* [29]. In this study, we report the development of a *G. mellonella* model for *Coccidioides posadasii*, which possesses a unique morphology compared to those of medically important fungi. We discuss its potential application in antifungal drug development and the identification of virulence factors. 

## 2. Materials and Methods

### 2.1. Coccidioides Strains, Culture Conditions, and Creation of Mutants

*C. posadasii* isolate C735 is a highly virulent clinical isolate. Arthroconidia (spores) were propagated in the saprobic phase on glucose–yeast extract (GYE) media (1% glucose, 0.5% yeast extract, and 1.5% agar) for 3 to 4 weeks at 30 °C in a biosafety level 3 laboratory, as previously reported [30]. All mutants were derived from the *C. posadasii* C735 parental isolate. The transformation of C735 by the *A. tumefaciens* strain LBA1100 harboring the Ti-DNA plasmid pCM41 (a gift from Dr. Chad Rappleye, Ohio State University) was conducted using a protocol reported by Marion et al., with some modification [31]. Briefly, germ tubes were prepared from *Coccidioides* arthroconidia as previously described [9]. Germ tubes were incubated with *A. tumefaciens* at a ratio of 1:1 in the presence of acetosyringone (AS, 200 µM) to improve transformation efficiency. The coculture was plated on an uncharged nylon membrane (Pall Biodyne A, 0.45 mm) and placed on a solid induction broth medium containing AS for 2 days at room temperature [32]. The co-culture was then dislodged and plated on GYE agar containing 200 μM cefotaxime to kill *A. tumefaciens* and 75 µg/mL of hygromycin for the initial selection of transformants. The subsequent isolation of homokaryotic mutants from initial transformants was conducted as described in our previous report [9]. In total, 105 Ti-DNA insertion mutants were created. Clones of each homokaryotic mutant were maintained on GYE containing 75 µg/mL of hygromycin at 30 °C for 3–4 weeks to produce arthroconidia. A challenge inoculum was prepared as previously reported [9]. In brief, arthroconidia were harvested and stored in PBS at a concentration >10^7^ spores/mL at 4 °C. Spore viability was evaluated via plating dilutions on GYE plates and by counting colony-forming units (CFUs) after 2–3 days of incubation. The spores were diluted to the indicated doses on the dates of the challenge experiments. For colony size, hyphal growth was measured each day throughout a 7-day time course from CpT13, CpT30, and C735 grown at 30 °C. 

### 2.2. Challenge of G. mellonella Larvae with Coccidioides Spores

*G. mellonella* larvae were purchased from Carolina^®^ (Item#143928). The larvae were maintained at 20 °C until use. One day prior to challenge, healthy *G. mellonella* larvae (0.2–0.4 g) with no signs of pigmentation were transferred to the BSL3 laboratory and acclimatized overnight at 37 °C. To challenge the larvae, the Petri dishes were transferred to an ice pad for 60 s to anesthetize and immobilize them for injection (Figure 1A). Individual larvae were positioned on a stand to secure the body for injection with a Hamilton repeating syringe into the hemocoel of the last left proleg. Larvae were injected with 10^3^ to 10^6^ spores per 10 μL suspended in PBS. Melanin pigment formation and the survival of the infected larvae were monitored daily for 7 days post-challenge. The melanization scores were assigned for each larva ranging from 0 to 5 point(s): 0 represented no melanization present; 1 represented one to three pigmented segments of larva; 2 represented four or more melanization segments; 3 represented partial light -gray pigmentation along the back segment; 4 represented visible intensified black spots across the whole light-gray body; and 5 represented intensified melanization covering the whole body or death. The fungal burden of *Coccidioides* was determined at 7 days post-challenge via the homogenization of the larvae in 1 mL of sterile PBS and by plating three serial dilutions of the homogenate on GYE agar plates containing 50 µg /mL of chloramphenicol.

### 2.3. Microscopic Analysis of Coccidioides Morphology in the Infected G. mellonella Larvae

The infected larvae were fixed in 10% buffered formalin at day 5 post-challenge, embedded in paraffin, and cut longitudinally with a microtome to prepare histology sections (5 µm). The tissue sections were stained with the Gomori methenamine silver stain. The processing of the slides and stains was carried out by UT Health San Antonio Histology and Immunohistochemistry Core. The slides were analyzed via light microscopy using a ZEISS Axioscope-5 microscope equipped with a ZEISS Axiocam-503 color camera. 

### 2.4. In Vivo Antifungal Drug Susceptibility Tests Using Galleria Larvae

Antifungal drugs, including clinically approved Amphotericin B (0.4 and 2.0 µg/dose; Millipore Sigma, St. Louis, MO, USA; A2942), and our newly identified Sanguinarine Chloride Hydrate (0.23 and 0.46 µg/dose; Millipore Sigma, S5890) and Closantel (20 and 40 µg/dose; Selleck Chemicals, Houston, TX, USA; S4106) were prepared in 10 µL of PBS and administrated via a hemocoel injection in the caudal proleg opposite to the *Coccidioides* inoculation site. The tested concentrations were determined by referencing in vivo data from the relevant human and veterinary literature, from which the dosages were allometrically scaled down to equivalent doses per body weight for the *Galleria* larvae in this study [33,34,35]. Mock control larvae were injected with PBS alone. After treatment, the larvae were returned to their Petri dishes and placed in a 37 °C incubator with 10% CO_2_ to maintain *Coccidioides* parasitic growth. Survival and melanin formation scores of the *Coccidioides*-infected larvae were determined for 7 days as described above.

### 2.5. Mouse Challenge, Body Weight Measurement, Survival, and Fungal Burden

C57BL/6 mice were purchased from Charles River Laboratories. Ten-week-old mice were challenged with a suspension of ~450 viable spores (CFUs; >4 × LD_100_) of C. *posadasii* C735 or the mutant strains in 50 μL of PBS via oropharyngeal aspiration. Mice were identified by ear punch, and individual body weights were obtained by weighing mice at 24 h intervals for 30 consecutive days post-challenge. CFUs in the lungs and spleen at 14 days post-challenge were determined via dilution plating on GYE agar plates containing 50 µg/mL of chloramphenicol as previously described [10]. The number of CFUs was expressed on a log scale and presented as a box plot for each group of 8 mice. Separate groups of mice were scored for survival for 30 days post-challenge. 

### 2.6. Profiling Morphology and Growth of Coccidioides Mutants via Imaging Flow Cytometry

Arthroconidia and spherules were prepared from the C735 parental isolate and *C. posadasii*-derived Ti-DNA insertion mutant strains or CpT, CpT13, and CpT30 mutants as previously described [10]. Cells were stained with Calcofluor white (CFW), a fluorescent dye targeting the chitin and β-1,4-glucan in the fungal cell wall, then fixed with 4% paraformaldehyde. Cell analysis was conducted using an ImageStreamX-MKII image flow cytometer equipped with a 40× objective lens as previously described [36]. Image data were acquired using the INSPIRE software and analyzed using IDEAS software (version 6.2, Millipore, St. Louis, MO, USA).

### 2.7. Ethics Statement

All animal experiments were conducted at the University of Texas at San Antonio (UTSA) in accordance with the Institutional Animal Care and Use Committee (IACUC) guidelines and in full compliance with the United States Animal Welfare Act (Public Law 98–198) and National Institute of Health (NIH) guidelines. The UTSA IACUC approved the animal protocol used in this study. The experiments were conducted in an Animal Biosafety Level-3 (ABSL3) laboratory accredited by the Association for Assessment and Accreditation of Laboratory Animal Care (AAALAC).

### 2.8. Statistical Analysis

The Log-rank (Mantel-Cox) test was used to compare the survival of *G. mellonella* larvae, C57BL/6, and BALB/c mice. A *p*-value of <0.05 was considered statistically significant. The Mann–Whitney U test was used to compare the body weight percent change in the mice infected with the mutants versus the parental strains. The Kruskal–Wallis test was used to assess the difference in fungal burden in both *G. mellonella* and mice.

## 3. Results

### 3.1. Establishment of an Insect Larva Model of Coccidioidomycosis

We first established a safe hemocoel injection method to inoculate *Galleria* larvae with virulent *Coccidioides* spores inside the BSL3 laboratory. Before inoculation, larvae (*n* = 20) were kept inside Petri dishes and then anesthetized on an ice pad for 60 s to induce immobility. Each larva was injected in the caudal left proleg with a dose of 10^3^ to 10^6^ C735 spores in 10 µL of PBS (Figure 1A). The infected larvae were placed back in their Petri dishes, secured with covers, and kept in a 37 °C incubator supplemented with 10% CO_2_ for 7 days. Both high CO_2_ and mammalian physical temperature conditions were required to maintain the parasitic growth of spherules. Mortality was observed in the group of larvae that were challenged with a dose of 10^6^ *Coccidioides* spores (Figure 1B). To curtail the range of LD50 further, we inoculated the larvae with an incrementally increased dose of *Coccidioides* spores from 1 × 10^5^ to 1 × 10^6^, as indicated in Figure 1C. Mortality was dose-dependent. The LD_50_ was between 4 and 6 × 10^5^ spores of *C. posadasii* C735 (Figure 1C). A positive relationship between coccidioidal challenge dose and melanization score, which peaked around 2 days post-challenge (dpc), is demonstrated in Figure 1D, E. Data showed that larvae received a higher number of coccidioidal spores, resulting in quicker and more abundant melanin synthesis and deposition.

### 3.2. Coccidioides Spores Propagated and Grew into Mixed Hyphae and Spherules Inside the Larvae 

A group of larvae were challenged with 5 × 10^5^ spores (CFUs) for fungal burden assays and the histological examination of *Coccidioides* growth and morphology in *G. mellonella* larvae. First, the fungal burden was remarkably reduced to 4 × 10^4^ CFUs at 2 dpc (Figure 2A). Steadily, it was increased to 6 × 10^5^ by day 7 post-challenge, indicating the growth and propagation of *Coccidioides*. Tissue sections showed nodule formation in the fat tissue under the outer cuticle layer of the larvae. Within the nodule, *Coccidioides* arthroconidia converted into hyphal fragments and spherules that had a cell size ranging from 15 to 50 µm in diameter (Figure 2B). Numerous spherules at various development and differentiation stages from small young (<10 µm; marked with blue arrows), segmenting (>20 µm; marked with white arrows), and endosporulating spherules (marked with red arrows) were visible (Figure 2B). Segmenting spherules were the dominant morphology while endosporulating spherules could be found in the nodule tissue. Those data revealed that *Coccidioides* could complete its parasitic cycle in *Galleria* larvae kept in a similar parasitic culture condition (37 °C and 10% CO_2_). Additionally, hypha-like cells (Figure 2B, marked with yellow arrows), which have been found inside subcutaneous granuloma of a murine model of dermal *Coccidioides* infection [37] and human chronic coccidioidal lung cavitation [38], were also present in the infected larval tissues.

### 3.3. Application of G. mellonella Model for Evaluation of Antifungal Drug Efficacy against Coccidioides Infection

We used Amphotericin B (AmB), a potent anti-*Coccidioides* drug, to test the suitability of the *Galleria* infection model for evaluating therapeutic efficacy. *Galleria* larvae were challenged with *C. posadasii.* C735 (5 × 10^5^ viable spores) and treated with AmB (0.4 or 2 µg per larva) via the same intra-hemocoel injection route on the opposite side of the inoculated proleg starting 2 hr post-challenge. The larvae received two additional doses of AmB at the same concentration on days 2 and 4 (Figure 3A). As shown in Figure 3B, injection with AmB alone did not cause any mortality, indicating that the drug is tolerable in these worms. All mock-treated and C735-challenged larvae (C735 + PBS group) succumbed to infection by day 5 or 6. In contrast, AmB treatment significantly increased survival (Figure 3B), supporting that the *Galleria* infection model could be applied for the in vivo evaluation of drug candidates against *Coccidioides* challenge. 

We tested this model using two drug candidates, Sanguinarine Chloride Hydrate (SANG) and Closantel (CLO), which were identified from our screening of a BROAD repurposing library against spherules (Sarah Sager et al., manuscript in preparation). Results suggest that SANG treatment is effective against coccidioidomycosis using the same AmB-treatment regimen (Figure 3A). The effective doses of SANG were 0.23 and 0.46 µg/day (Figure 3C), while CLO was toxic at a dose of 40 µg/day. It had no therapeutic benefit in this insect model of coccidioidomycosis (Figure 3D).

### 3.4. Galleria Larva Model Can Be Used to Screen Attenuated Mutants of Coccidioides

We generated a small set of hygromycin-resistant *Coccidioides posadasii* mutants (105 clones) using an *A. tumefaciens*-mediated random Ti-DNA insertion. Among those mutants, 24 clones showed normal saprobic and hyphal growth and produced arthroconidia at a rate similar to that of the parental C735 strain. We challenged *Galleria* larvae (*n* = 10) with 5 × 10^5^ viable spores isolated from 24 tested mutants. Larvae challenged with an equal number of spores isolated from C735 served as a control. CpT13 and CpT30 mutants were significantly attenuated in pathogenicity, resulting in elevated survival at 80% and 90%, respectively, compared to that of larvae challenged with the parental C735 strain (Figure 4A). CpT13-infected larvae demonstrated significantly reduced melanization, and CpT30 also showed a similar trend of decreased melanization (Figure 4B). We further characterized the CpT13 and CpT30 mutants using a fungal burden assay. Results showed that both mutants significantly reduced fungal burdens by >150 fold compared to those of the parental strain (2 × 10^2^ versus 3 × 10^4^; Figure 4C). Other tested mutants including CpT1, CpT4, CpT7, CpT36, CpT44, and CpT47 did not differ from C735 (WT) in pathogenicity including survival, melanization, and fungal burden (Figure 4A–C).

### 3.5. CpT13 and CpT30 Mutants Displayed Normal Saprobic Growth but Reduced Parasitic Cell Sizes

The CpT13 and CpT30 mutants exhibited hyphae and arthroconidia of the same size and rate as those of the parental strain, *C. posadasii* C735. Both CpT13 and CpT30 formed hyphal mats with a colony diameter comparable to that of the *C. posadasii* C735 strain as shown at 3 and 6 days post-inoculation and were quantified throughout a 7-day culture (Figure 5A). Microscopic examination of the hyphae of the two mutants showed a morphology similar to that of the parental strain, with comparable thickness and hyphal branching (Figure 5B). Moreover, there was no significant difference in arthroconidium size. CpT13 and CpT30 arthroconidia can undergo isotropic growth into spherules in a chemically defined Converse medium at 39 °C with 10% CO_2._ Imaging flow analysis showed that 82.8% of C735 spherules were enlarged with a surface area larger than 250 units, while the two mutants had a significantly reduced spherule size compared to that of parental strain (7.2% and 7.23%, respectively), as shown in Figure 5C. The gated spherule population from the parental strain averaged over 10 μm in size, while the mutant strains had a reduced spherule size, as shown in Figure 5D. These data suggest that the two mutants exhibit normal saprobic growth but are deficient in parasitic growth, which may explain their reduced virulence.

### 3.6. CpT13 and CpT30 Reduced Virulence in Two Murine Models of Pulmonary Coccidioidomycosis

We further validate these findings using two highly susceptible murine models of pulmonary coccidioidomycosis. BALB/c mice were used to evaluate survival, while C57BL/6 mice served as a model for fungal burden. To assess the relative virulence of *C. posadasii* mutants compared to that of the parental isolate (C735), each of the five groups of BALB/c mice (*n* = 8) were challenged with an equivalent concentration of viable spores (450 spores) obtained from each mutant (CpT4, CpT13, CpT30, and CpT47), as well as the C735 parental isolate that served as a control. Results revealed that CpT13 and CpT30 significantly maintained survival after 30 days post-challenge, while mice challenged with the CpT4, CpT47, and parental *C. posadasii* C735 strains succumbed to fatal disease within 19 days (Figure 6A). We further compared the fungal burden of C57BL/6 mice (*n* = 8) that were challenged with an equal dose (465 spores) of CpT13, CpT30, and C735 strains. Similarly, the fungal burdens were significantly reduced in the lungs and the spleen of CpT13- and CpT30-challenged mice at day 14 post challenge, compared to those of mice challenged with C735 (Figure 6B). Notably, CpT30-challenged mice had significantly lower CFUs in the lungs compared to the CpT13-challenged mice, and neither of these disseminated toward the spleen (Figure 6B). These data showed that attenuated mutants identified using the larva model also displayed reduced virulence in both murine models of pulmonary coccidioidomycosis. These data strongly indicate that the larva model is suitable as a screening tool for identifying and characterizing attenuated *Coccidioides* mutants.

## 4. Discussion

The *Galleria* larva model has gained traction as an alternative animal model for several infectious diseases. Larvae exhibit a humoral immune response to microbial infections and the activation of specific immune cells that is similar to the innate immune response in murine models [39]. *G. mellonella* innate systems include pattern recognition receptors to recognize microbial invasion and phagocytic cells like plasmatocytes and hemocytes to prevent colonization and infection (reviewed by Smith and Casadevall [40]). Reported data indicate that fungal virulence factors have comparable immune function roles in mammalian and *G. mellonella* hosts, suggesting that *G. mellonella* studies can be translatable to mammals [40]. *Galleria* larvae can grow under versatile conditions, making them an excellent model for research on *Coccidioides*. These fungal species display a unique dimorphic lifestyle compared to that of all medically important fungal pathogens, as they are the only known fungal pathogens to form multinucleated spherules in mammalian tissues. In vitro, cultures of *Coccidioides* are incubated at 37–39 °C in 10% CO_2_ to mimic mammalian infection conditions and maintain spherule growth. In *Galleria* under these conditions, our study revealed the conversion of *Coccidioides* arthroconidia into spherules, a critical event for coccidioidal pathogenicity in mammalian hosts, and the simulation of its parasitic growth in mammals. 

Our current studies are limited to evaluating the *Coccidioides posadasii* mutant library and the parental isolate C735. Future studies have the potential to explore virulence in both *C. posadasii* and *C. immitis* isolates*,* which are phylogenetically related, showing a 4–5% difference in genomic sequences and sharing indistinguishable disease manifestations, albeit with variations in geographic distribution. *C. immitis* has been historically present in arid desert regions of Central and Southern California, while *C. posadasii* is located outside of California while also being in Nevada, Arizona, New Mexico, west Texas, Mexico*,* and Central and South America. Despite *C. posadasii* and *C. immitis* sharing major virulence determinants and structural genes, there are variations in amino acid sequences of the translated proteins, as revealed in genomic comparisons. Recent studies using a recombinant subunit vaccine (GCP-rCpa1) have demonstrated for the first time that a vaccine created based on the *C. posadasii* genome can be cross-protective in *C. immitis* pulmonary challenge in a murine model [41].

*Coccidioides* species are nonspecific pathogens that can cause disease in many mammals, and several experimental animal models have been applied to study their pathogenesis, chemotherapeutic efficacies, and vaccine-induced protective immunity [18]. In a previous study using *Galleria*, the pathogenicity of a Silveira strain of *Coccidioides posadasii* maintained in the spherule/endospore phase (S/E) was evaluated. The study demonstrated a one hundred percent mortality rate by 4 days post-infection using 1 × 10^6^ spherules [42]. Since the parasitic spherule phase can contain 100–1000 endospores and quantification post-mortem was not conducted, it remains unclear what the fungal burden required for LD_100_ was. Further investigation into the melanization score and fungal burden could provide valuable insights into the S/E strain. The LD_50_ and LD_100_ of the *Galleria* larva model of coccidioidomycosis are surprisingly high at 0.5–1.0 × 10^6^ *Coccidioides* spores, compared to those of susceptible mice, which are <100 spores. The lethal doses of opportunistic pathogens like *Candida albicans* and *Cryptococcus neoformans* strain H99 are reported to be 1–2 × 10^5^ cells. The lethal dose for *Coccidioides* infection in *Galleria* appears comparable to those of fungal pathogens. 

The larva model was applied to validate newly identified anti-*Coccidioides* compounds for drug development and to screen *Coccidioides* mutants with a disruption of virulence factors. These data further support the suitability of the *Galleria* larva model for studying *Coccidioides* infection, especially as a validation tool for large quantitative samples in drug screening and virulence testing. The application of the *Galleria* model to evaluate antifungal activity has been widely reported [22,43,44]. Cruz et al. reported the antifungal activity of thiazolyl hydrazone compounds against *Candida albicans* identified in the *Galleria* model and then confirmed their efficacy in a murine model [45]. In this study, we demonstrated anti-*Coccidioides* activity for the clinical drug AmB and a candidate compound, SANG, in our established *Galleria* model of coccidioidomycosis. In our study, the consistent efficacy observed at both dosages, 0.23 µg and 0.46 µg, for SANG provides intriguing insights into the larvae’s pharmacodynamics. The similarity in effectiveness at these distinct doses suggests the possibility of achieving maximum therapeutic benefits at a lower dosage. This phenomenon may indicate that the drug operates at its plateau within this range, implying that further increases in dosage beyond 0.23 µg would not confer additional advantages. Meanwhile, our second candidate compound, CLO, did not improve the outcome of coccidioidal infection. These results allow us to prioritize candidate drugs for further evaluation in the murine model of coccidioidal infection. 

The *Galleria* model also provides an excellent in vivo platform to evaluate coccidioidal virulence factors. As shown in Figure 4 and Figure 6, we were able to identify two highly attenuated C735 mutant strains from a small collection of Ti-DNA insertion mutants. The virulence of C735 mutants seems to be highly correlated between the larva and murine models. Both *Galleria* larvae and mice are less susceptible to the attenuated CpT13 and CpT30 strains than they are to the C735 strain, showing reduced mortality and morbidity (melanization in larvae or weight loss in mice). The Ti-DNA insertion site of the CpT30 mutant was mapped to a *Coccidioides* hypothetical gene (XM_003065484.1; data not shown). Genetic and microbiological characterization for CpT13 and CpT30 is underway to identify the mechanisms responsible for their attenuation in pathogenesis.

In summary, we have developed a *Galleria* model of coccidioidomycosis. Collective results from this study indicate that this mini-host insect model is a valuable alternative for studying coccidioidal pathogenesis. The application of this larva model was successfully demonstrated for anti-coccidioidal drug evaluation and the identification of attenuated coccidioidal mutants in this report. Moreover, this established *Galleria* model may be of great value to additional coccidioidal studies including, but not limited to, those of host–pathogen interaction (fungal pathogenicity–host innate response), virulence factor characterization, and microbial co-infection. 

## Figures and Tables

**Figure 1 jof-10-00131-f001:**
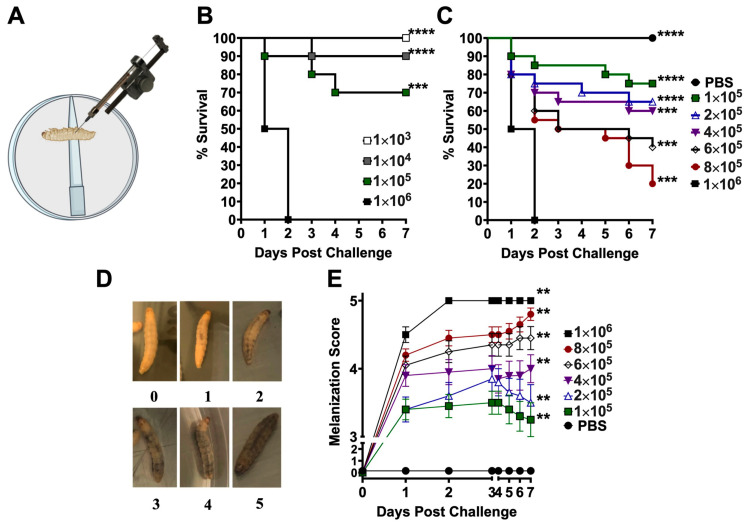
*Galleria mellonella* model of coccidioidomycosis. An illustration of the hemocoel injection method (**A**). Survival plots of groups of *Galleria* larvae (*n* = 20) that were challenged with an indicated dose of viable arthroconidia prepared from the virulent *C. posadasii* C735 isolate (**B**,**C**). Larvae incubated at 37 °C with 10% CO_2_ and monitored daily for survival (**** *p* < 0.0001, *** *p* < 0.001), via the log-rank test. (**D**) Representative larvae showing the degree of melanization ranging from scores of 0 to 5: 0, no melanization present; 1, one to three pigmented segments of larvae; 2, four or more melanization segments; 3, partial light-gray pigmentation along the back and tail; 4, visible intensified black spots across the whole light-gray body; and 5, intensified melanization covering the whole body. (**E**) Plots of melanization scores for each challenged group and controls for a period of 7 days (** *p* < 0.05 via Mann–Whitney test).

**Figure 2 jof-10-00131-f002:**
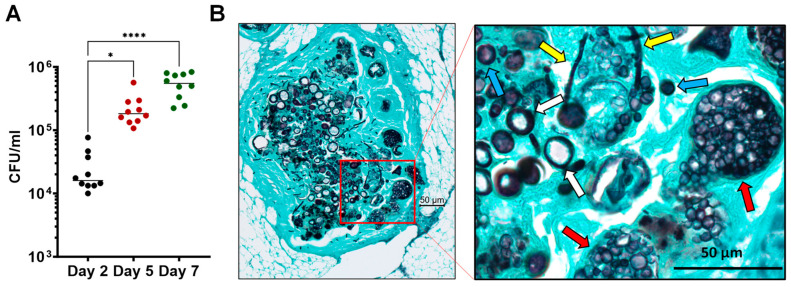
Parasitic growth and propagation of *Coccidioides posadasii* in *G. mellonella* larvae. *Galleria* larvae (*n* = 10 per time point) were challenged with 5 × 10^5^ *C.posadasii* C735 spores. (**A**) Fungal burdens measured on day 2, 5, and 7 post-challenge. * *p* < 0.05 and **** *p* < 0.0001 via the Kruskal–Wallis test. (**B**) Micrographs of *G. mellonella* larvae infected with *Coccidioides* at 5 dpc. *Galleria* larvae sections were stained with Gomori methenamine silver stain (GMS). The left panel shows nodule formation in the fat tissue under the outer cuticle layer of the larvae. The enlarged insert on the right panel shows various developmental morphologies and stages of *Coccidioides*; the red arrows indicate the release of endospores from a ruptured spherule, the yellow arrows indicate hypha-like cells, the white arrows indicate a segmenting spherule, and the blue arrows indicate small spherules.

**Figure 3 jof-10-00131-f003:**
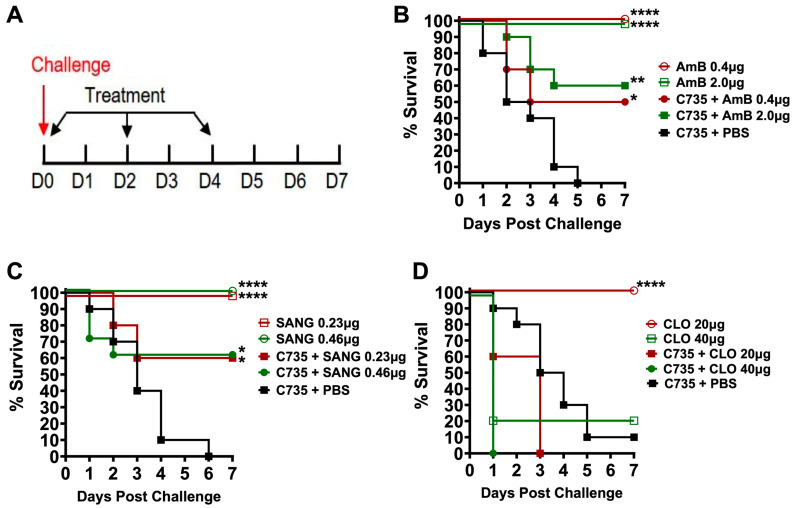
Application of *Galleria* larva model of coccidioidomycosis for antifungal discovery. (**A**) Timeline and treatment: groups of larvae (*n* = 10) challenged with 5 × 10^5^ spores of C735, rested for 2 h, and then treated with Amphotericin B (AmB) (**B**), Sanguinarine (SANG) (**C**), or Closantel (CLO) (**D**) at indicated doses. Treatment with PBS served as a control. The larvae received two additional doses of each drug at 2- and 4-days post-challenge. Unchallenged larvae that received each drug alone were used to assess drug toxicity. **** *p* < 0.0001, ** *p* < 0.01 and * *p* < 0.05 via the log-rank test.

**Figure 4 jof-10-00131-f004:**
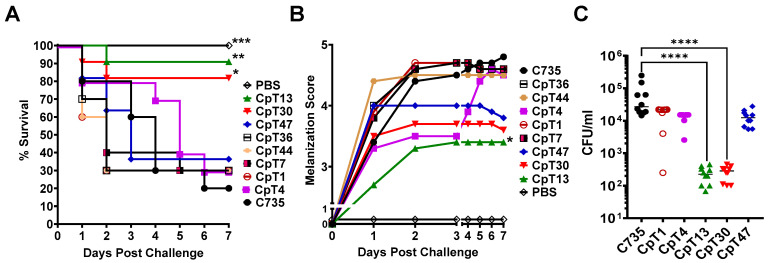
Screening attenuated mutants of *Coccidioides posadasii* using the larva model. Groups of *Galleria* larvae (*n* = 10) were challenged with 5 × 10^5^ viable spores prepared from parental *C. posadasii* C735 and indicated Ti-DNA insertion mutant strains (CpTs). The larvae were incubated at 37 °C. and monitored daily for (**A**) survival rate of mutants were compared to C735 by log-rank test Mantel-Cox *** *p* < 0.001, ** *p* < 0.01, * *p* < 0.05). (**B**) mean melanization scores for a period of 7 days. (**C**) Fungal burden at day 7 post-challenge were compared to C735 via one-way ANOVA with Kruskal–Wallis test (**** *p* < 0.0001).

**Figure 5 jof-10-00131-f005:**
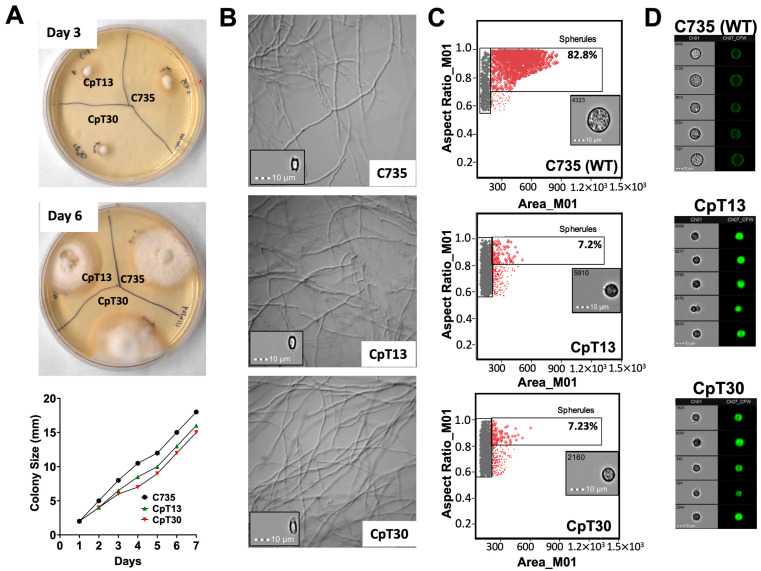
Growth of the CpT13 and CpT30 attenuated mutants compared to the C735 parental isolate on GYE and Converse media. (**A**) Colony size was measured (in mm) for each strain grown on a GYE plate at 30 °C for a 7-day period. Representative photographs of colonies were taken at 3 and 6 days post-inoculation. Their growth curves were comparable. (**B**) Micrographs of hyphae isolated from the 7-day culture of the two mutants and the parental strains in GYE medium showing similar morphologies. They also formed visually identical arthroconidia with similar sizes and morphologies. (**C**) Debris and small fungal cells (gray) and large gated spherules (red) were gated from 4-day cultures of the mutants and parental strains in the Converse medium using image flow cytometry as described in the Materials and Methods section. The insert image in each panel is of a representative spherule of each strain. (**D**) Spherule images labeled with Calcofluor white selected from gated spherule populations of each strain. The images are representative of spherules of an average size in each population.

**Figure 6 jof-10-00131-f006:**
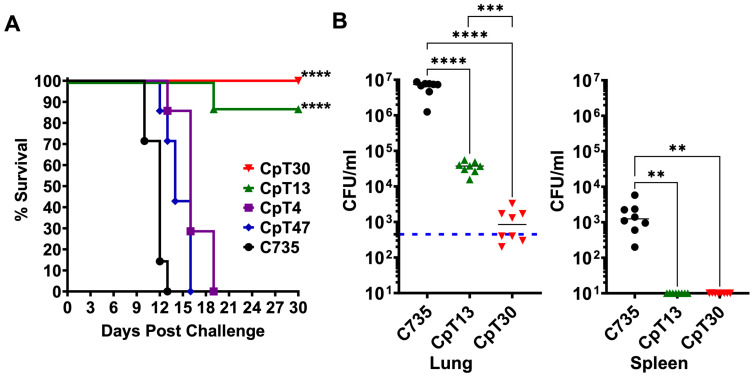
Evaluation of virulence of *Coccidioides posadasii* Ti-DNA insertion mutants using a mouse model of coccidioidomycosis. Groups of BALB/c mice (*n* = 7–8) were challenged with 450 viable spores prepared from parental Cp C735 and indicated Ti-DNA insertion mutant strains (CpTs). (**A**) Mice monitored for daily survival for a period of 30 days. (**B**) Fungal burdens in the lungs and the spleen determined at day 14 post-challenge in C57BL/6 mice. The blue dashed line represents the challenge dose. The survival rate, **** *p* < 0.0001 via the log-rank (Mantel–Cox) test compared to that of C735. The fungal burden, **** *p* < 0.0001, *** *p* < 0.001 and ** *p* < 0.01, was evaluated via one-way ANOVA with the Kruskal–Wallis test.

## Data Availability

Data are contained within this article.

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
