# Peer review of "Galleria mellonella* Model of Coccidioidomycosis for Drug Susceptibility Tests and Virulence Factor Identification"

_jof, 2024, doi:10.3390/jof10020131_

Round 1

Reviewer 1 Report

Comments and Suggestions for Authors

This manuscript presents a new experimental animal model of coccidioidomycosis in an invertebrate of the genus Galleria. The authors demonstrated that this animal model is useful to the study of Coccidioides infection, its pathogenesis, the evolution of the infection with less virulent Coccidioides strains, and the action of new antifungal drugs in this experimental model. This study opens new ways to research this endemic mycosis in the American continent. This study was well-planned, very well executed and well presented.

1. The study aims to study the utility of a new experimental model of coccidioidomycosis in an invertebrate host.

2. This study allowed the authors to demonstrate the usefulness of this model in studies of pathogenesis, antifungal action, and experimental disease with less virulent strains of Coccidioides sp.

3. This experimental model was used for the first time in this mycosis and the findings of this research open new avenues for studies such as vaccination with less virulent strains.

4. The main advantage of the use of invertebrates is their lower cost, these larvae had already been successfully studied in other mycoses but in coccidioidomycosis which is an endemic disease in America and mainly in the USA whose area is spreading.

5. The conclusions are based on the findings of the studies carried out.

6. The cited bibliography is related to the subject under study and is correct and sufficient.

7. The tables are clear, and necessary and help to understand the text.

Author Response

We appreciate the reviewer’s time to review and provide kind comments. We have made minor modifications based on other suggestions by reviewers to improve the manuscript further. We thank you for your time and feedback.

Reviewer 2 Report

Comments and Suggestions for Authors

The Galleria mellonella Model of Coccidioidomycosis for Drug Susceptibility Tests and Virulence Factor Identification manuscript is well built. The focus is to study Galleria as a model to evaluate virulence and its application to drug tests in an important dimorphic fungus, the Coccidioides genus. The work can be published after revision—a well-written manuscript with relevant data.

Abstract

The summary presents the primary data obtained in the work, but it remains to mention which species of Coccidioides will be used in the study.

L 15- “Due to an effective innate defense system of Galleria, the (LD100) is between 0.5-1.0 x 106 viable Coccidioides spores for all clinical isolates”.

This mention suggests that the immunological response was evaluated, but the manuscript did not consider this aspect.

This mention suggests several clinical isolates were used, but a single strain of C. posadasii isolate C735 was used in the text.  

L-20- coccidioidal mutants. Precisa mencionar que são mutantes da cepa de C. posadasii isolate C735.

A virulence factor study is mentioned, but this data does not appear.

Introduction

The text is excellent but needs to be included in the objective that C. posadasii clinical strain and mutants will be used.

The text must be updated, mainly indicating the dimorphic fungi studied with the model.

Cite reference DOI: 10.3390/jof8050455—the first study with the Galleria model in Coccidioides.

Results

In Figure 5, the plates with the fungus growing in the mycelial phase are very bad and do not show much difference. It would be better to put microscopic aspects.

L- 271 “as well as the C735 wild type (450 spores)”. In L 11, “Studies of coccidioid virulence and pathogenesis are mostly conducted using murine models with low lethal doses (LD100 <100 spores). This statement refers to C. posadasii. Or C. immitis? The manuscript showed different concentrations. 

Author Response

We sincerely thank the reviewer for providing invaluable feedback and insightful comments on our manuscript. We have made the following modifications based on the reviewer’s comments.

The Galleria mellonella Model of Coccidioidomycosis for Drug Susceptibility Tests and Virulence Factor Identification manuscript is well built. The focus is to study Galleria as a model to evaluate virulence and its application to drug tests in an important dimorphic fungus, the Coccidioides genus. The work can be published after revision—a well-written manuscript with relevant data.

Abstract

  • The summary presents the primary data obtained in the work, but it remains to mention which species of Coccidioides will be used in the study.

Thank you for bringing this to our attention. We have modified the abstract to illustrate the work conducted in C. posadasii and the mutant library derived from this species, as described in lines 14-16.

  • L 15- “Due to an effective innate defense system of Galleria, the (LD100) is between 0.5-1.0 x 106 viable Coccidioides spores for all clinical isolates”. This mention suggests that the immunological response was evaluated, but the manuscript did not consider this aspect.

We have omitted the immunological response of Galleria; as the reviewer pointed out, this was not the scope of the manuscript, and we have modified lines 11-14.

  • This mention suggests several clinical isolates were used, but a single strain of posadasii isolate C735 was used in the text.  

L-20- coccidioidal mutants. Precisa mencionar que são mutantes da cepa de C. posadasii isolate C735.

A virulence factor study is mentioned, but this data does not appear.

We thank you for the helpful feedback. We have specified the species used for the mutant library (line 18-19) and have added additional information to Figure 5 and corresponding text (lines 254-268) to further evaluate virulence factors based on the saprobic vs parasitic phase of the mutants tested.

Introduction

The text is excellent but needs to be included in the objective that C. posadasii clinical strain and mutants will be used.

Thank you for bringing this to our attention. We have further specified the species used in lines 75-76.

  • The text must be updated, mainly indicating the dimorphic fungi studied with the model.

Cite reference DOI: 10.3390/jof8050455—the first study with the Galleria model in Coccidioides.

We have incorporated the first study using the Galleria model in Coccidioides under the discussion in line 329-336.

Results

  • In Figure 5, the plates with the fungus growing in the mycelial phase are very bad and do not show much difference. It would be better to put microscopic aspects.

 We agree that the initial Figure 5 does not represent variations in morphology well; we have modified Figure 5 and have edited the text in lines 254-268. 

  • L- 271 “as well as the C735 wild type (450 spores)”. In L 11, “Studies of coccidioid virulence and pathogenesis are mostly conducted using murine models with low lethal doses (LD100 <100 spores). This statement refers to C. posadasii. Or C. immitis? The manuscript showed different concentrations. 

We have made modifications to the text discussing the rationale of using a higher inoculum for the survival and fungal burden studies to demonstrate differences in virulence between C735 parental isolate and avirulent mutant strains in lines 278-282.
